# Discovery of a *MUC3B* gene reconstructs the membrane mucin gene cluster on human chromosome 7

Tiange Lang[1], Thaher Pelaseyed[2]*

1 Big Data Decision Institution, Jinan University, Tianhe, Guangzhou, China, 2 Department of Medical Biochemistry and Cell Biology, Institute of Biomedicine, University of Gothenburg, Gothenburg, Sweden

* thaher.pelaseyed@medkem.gu.se

## Abstract

Human tissue surfaces are coated with mucins, a family of macromolecular sugar-laden proteins serving diverse functions from lubrication to the formation of selective biochemical barriers against harmful microorganisms and molecules. Membrane mucins are a distinct group of mucins that are attached to epithelial cell surfaces where they create a dense glycocalyx facing the extracellular environment. All mucin proteins carry long stretches of tandemly repeated sequences that undergo extensive O-linked glycosylation to form linear mucin domains. However, the repetitive nature of mucin domains makes them prone to recombination and renders their genetic sequences particularly difficult to read with standard sequencing technologies. As a result, human mucin genes suffer from significant sequence gaps that have hampered the investigation of gene function in health and disease. Here we leveraged a recent human genome assembly to characterize a previously unmapped *MUC3B* gene located at the q22 locus on chromosome 7, within a cluster of four structurally related membrane mucin genes that we name the MUC3 cluster. We found that *MUC3B* shares high sequence identity with the known *MUC3A* gene and that the two genes are governed by evolutionarily conserved regulatory elements. Furthermore, we show that *MUC3A*, *MUC3B*, *MUC12*, *and MUC17* in the human MUC3 cluster are expressed in intestinal epithelial cells (IECs). Our results complete existing genetic gaps in the MUC3 cluster which is a conserved genetic unit in vertebrates. We anticipate our results to be the starting point for the detection of disease-associated polymorphisms in the human MUC3 cluster. Moreover, our study provides the basis for the exploration of intestinal mucin gene function in widely used experimental models such as human intestinal organoids and genetic mouse models.

## Introduction

The first draft of the human genome published twenty years ago offered a unique opportunity to decipher the causal relationship between genetic sequence, gene function, and disease biology [1, 2]. But reading and measuring repetitive genomic elements remains a major

**Data Availability Statement:** All relevant data are within the manuscript and its Supporting Information files. Gapless mucin gene sequences at human chr 7q22 will be publicaly available via

the Mucin database (http://www.medkem.gu.se/
mucinbiology/databases/index.html.

**Funding:** TP was supported by - Grant S17-0005,
Swedish Society for Medical Research, https://
www.ssmf.se - Grants 5U01AI095542-08-WU-19-
95 and 5U01AI095542-09-WU-20-77, National
Institutes of Health, https://www.niaid.nih.gov -
Grants FT2017-0002, UPD2018-0065, and
WUP2017-0005, Wenner-Gren Foundations,
https://www.swgc.org/ - Grant JS2017-0003,
Jeansson Foundations, http://jeanssonsstiftelser.
se/en/ - Grant M17-0062, Åke Wiberg Foundation,
https://ake-wiberg.se/ The funders had no role in
study design, data collection and analysis, decision
to publish, or preparation of the manuscript.

**Competing interests:** The authors have declared
that no competing interests exist.

technological challenge that has left the human genome riddled with significant sequence
gaps. Mucin (*MUC*) genes are characterized by subexonic repeats, consisting of multiple
repeated short DNA sequences within coding exons. The resulting tandemly repeated
sequences encode extended domains that are rich in proline, threonine, and serine (PTS) residues [3]. Mucin-type tandem repeats undergo O-linked glycosylation on serines and threonines to form densely O-glycosylated linear mucin domains [4]. The number and sequence
identity of tandem repeats vary between *MUC* genes and are further confounded by considerable length polymorphism between individuals, resulting in variable number of tandem
repeats (VNTRs). VNTRs present a major challenge in analyzing mucin sequences since their
repetitive nature and size in several kilobases cause intrinsic instabilities that are difficult to
maintain in bacterial artificial chromosomes. Consequently, mucin gene VNTRs are underrepresented in the human genome assembly [5] and continue to hamper efforts to investigate
*MUC* gene function.

Mucins are an ancient family of proteins in the animal kingdom. The earliest mucin
genes appeared 700–800 million years ago in primitive marine metazoans such as sea anemones, sponges, and jelly combs and have since expanded to all branches of the tree of life [6,
7]. Currently, the human mucin family consists of secreted gel-forming mucins (MUC2,
MUC5B, MUC5AC, MUC6, MUC7, and MUC20) and a distinct subfamily of membrane
mucins (MUC1, MUC3, MUC4, MUC12, MUC13, MUC15, MUC16, MUC17, MUC21,
and MUC22) that are inserted into cell membranes via a transmembrane domain [8]. Membrane mucins are single-pass type I transmembrane proteins that are guided to the secretory
pathway via an N-terminal signal sequence. In the endoplasmic reticulum, membrane
mucins undergo N-linked glycosylation and folding, which in most cases requires a strain-dependent autocatalytic cleavage at a Sea urchin sperm protein, Enterokinase and Agrin
(SEA) domain [9]. The cleaved protein fragments remain non-covalently attached at the
SEA domain as the mucin protein transits to the Golgi apparatus for O-linked glycosylation.
Consequently, the mature SEA-type membrane mucin reaches the plasma membrane as a
heterodimer with a glycosylated extracellular N-terminal subunit that remains non-covalently linked to a membrane-attached C-terminal subunit. The evolutionary origins of the
SEA domain date back to single-celled eukaryotes while SEA-type membrane mucins
emerged in vertebrates [3, 10]. The SEA domain is a mechanosensor that undergoes conformational unfolding in response to mechanical tension, but its ultimate biological function
remains elusive [11].

In humans, the SEA-type membrane mucin genes *MUC3*, *MUC12*, and *MUC17* map to the
chromosomal locus 7q22. The three genes are arranged in a *MUC3-MUC12-MUC17* cluster
(hereafter called MUC3 cluster), and flanked by *ACHE* upstream of *MUC3*, and *TRIM56 and
SERPINE1* downstream of *MUC17* (Fig 1A). The stereotypic *ACHE-MUC3-MUC12-MUC17-
TRIM56- SERPINE1* unit is highly conserved in vertebrates. *Mus musculus* carries a MUC3
cluster on chromosome 5, where three membrane mucin genes are flanked by *Ache* and
*Trim56-Serpine1*. Notably, the *Muc3* gene in *M. musculus* maps directly upstream of *Trim56*
and shares 43% sequence identity with human *MUC17*, but only 28% identity with human
*MUC3*, indicating that murine *Muc3* is a homolog of human *MUC17* while the murine homologs for *MUC3* and *MUC12* are poorly defined [3]. The nonmammalian vertebrate *Xenopus
tropicalis* carries seven genes encoding SEA-type membrane mucins, of which three are
arranged in tandem on chromosome 3 followed by a homolog of human *SERPINE1*, suggesting that the MUC3 cluster first emerged in amphibians [3].

The current human genome assembly GRCh38.p13 is estimated to contain unsolved gaps
corresponding to nearly 150 million base pairs (Mbp) [12], which we postulate underlie the
lack of complete sequences for human *MUC* genes in general and the MUC3 cluster in

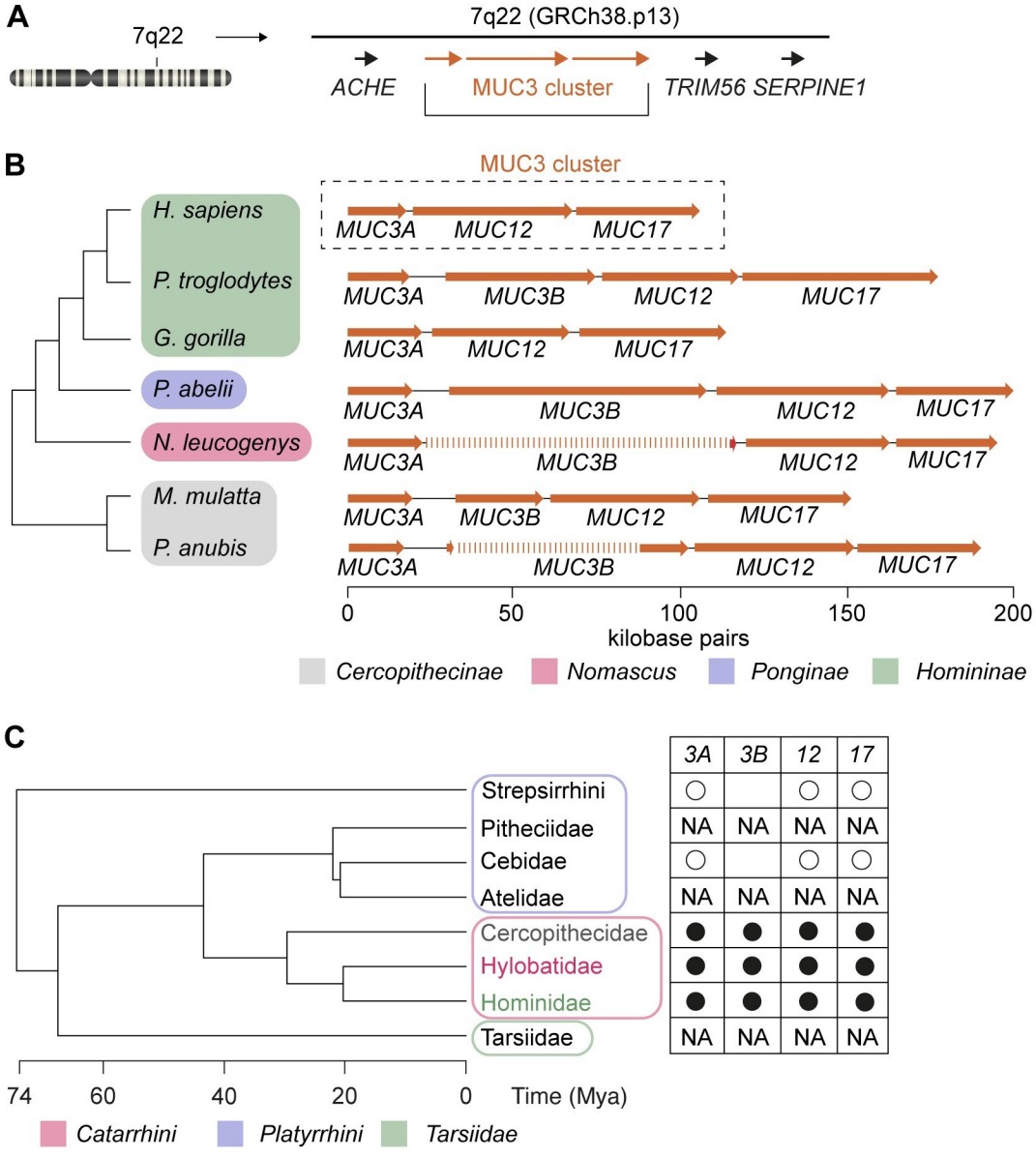

**Fig 1. Conservation of MUC3 cluster in Cercopithecoid and Hominoid superfamilies.** (A) The MUC3 cluster at locus q22 in human chromosome 7 in the GRCh38.p13 assembly is flanked by genes *ACHE*, *TRIM56*, and *SERPINE1*. (B) Members of the Cercopithecoid and Hominoid superfamilies, except for *H. sapiens* and *G. gorilla*, carry a MUC3 cluster consisting of *MUC3A*, *MUC3B*, *MUC12*, and *MUC17* genes. (C) Presence of the *MUC3B* gene in MUC3 cluster in Catarrhini parvorder (filled black circles). Open circles indicated the presence of *MUC3*, *MUC12*, and *MUC17* genes in the Platyrrhini parvorder. NA indicates a lack of sufficient sequence information for the detection of MUC3 cluster genes.

particular. In this work, we take advantage of the most recent T2T-CHM13 assembly of the human genome [12] to provide evidence for the existence of a human *MUC3B* gene. We also demonstrate that *MUC3A* and *MUC3B* genes are conserved in late hominoids such as the chimpanzee as well as Old World monkeys. Finally, by exploring published RNA-sequencing data sets, and applying quantitative gene expression analysis in human tissues, we show that *MUC3A* and *MUC3B* expression is limited to IECs.

## Material and methods

### Recruitment of patients and sample collection

Patients ≥18 years who were referred to Sahlgrenska University Hospital (Gothenburg, Sweden) for colonoscopy, were eligible for inclusion and subject to the provision of written informed consent. Patients with macroscopic/microscopic evidence of ileocolonic pathology other than Inflammatory bowel disease were excluded. Eight biopsies were obtained from the terminal ileum of each patient. The study protocol was approved by the regional ethics committee (Ethical permit #2020–03196) and complied with the Declaration of Helsinki.

### Phylogenetic data

Phylogenetic trees and molecular time estimates were extracted from TimeTree [6, 13].

### Sequence alignments

Local sequence similarity search and identity measurements of *MUC* genes were performed using NCBI BLAST [14]. Primer specificity was analyzed using primer BLAST [15]. Multiple sequence alignment of *MUC* gene and protein homologs was conducted using CLUSTALW [16]. Perl scripts were used for all data extraction (see supplementary methods in S1 File). Promotor regions -1 kb from the transcription start site of *MUC3A* and *MUC3B* in Cercopithecoid and Hominoid superfamilies were aligned using Multiple Alignment using Fast Fourier Transform (MAFFT) high-speed multiple sequence alignment tool [17].

### Generation of dot plots for pairwise sequence alignment and sequence logo representations

Dot plots representing pairwise sequence alignments were generated using Genome Pair Rapid Dotter (GEPARD) version 1.40 [18]. Sequence logos of perfect tandem repeats were generated using WebLogo3 [19].

### Mapping of DNase-seq and ChIP-seq data to the human genome

DNase hypersensitive sequences upstream of *MUC3A* in GRCh38.p13 and Chromatin immunoprecipitation (ChIP) sequencing of the human small intestine, colon, and stomach samples are summarized in S1 Table [20]. Graphical representation of epigenetic signatures was prepared by aggregating multiple segment-sorted tracks using the Matplot function in Washington University Epigenome Browser v53.5.0 [21].

### Single-cell expression of transcription factors

Expression profiles for transcription factors were extracted from the following data sets available at Single Cell Portal (Broad institute): single-cell transcriptome analysis of human small intestine (GSE148829) [22], human colon (GSE178341) [23], and mouse small intestine (GSE92332) [24].

### Mapping of RNA-sequencing data to T2T-CHM13 human genome assembly

The T2T-CHM13 human genome assembly was downloaded from NCBI BioProject PRJNA559484. Fastq-dump was used to obtain RNA-sequencing reads. Burrows-Wheeler Aligner (BWA) software package [25] was used to align RNA-sequencing reads to exonic sequences of genes belonging to the MUC3 cluster. Perl scripts were used to perform quality

control and measure read number (see supplementary methods in S1 File). The following publicly available data sets were used to determine *MUC* gene expression in human tissues: single-cell transcriptome analysis of human ileum, colon, rectum (GSE125970) [26], human liver (GSE124395) [27], and human kidney (GSE131685) [27]. Gene expression of individual *MUC* genes in the MUC3 cluster was calculated as transcripts per million (TPM) as previously described [28].

### RNA extraction from human ileum, cDNA synthesis, and RT-qPCR

RNA from human ileal biopsies was extracted using RNeasy Mini Kit (Qiagen). 500 ng of RNA was reverse transcribed to cDNA with TaqMan Reverse Transcription kit (#N8080234, Applied Biosystems), using 2.5 μM random primers and the cycling parameters 25.0˚C for 10 min, 37.0˚C for 30 min, and 95.0˚C for 5 min. 750 ng of cDNA was used for downstream reverse transcription quantitative PCR (RT-qPCR) with 0.3 μM *MUC3A*-specific primers (forward 5′-TGGGGGTCAGTGGGATGGCCTCAAA-3′; reverse 5′-CACGTGGGACCGCTCGTC TCC) or *MUC3B*-specific primers (forward 5′-CGGGGGCCAGTGGGATGGCCTCAAG-3′; reverse 5′-CACGCGGGACCGCTCGTCTCT-3′) using SsoFast EvaGreen Supermix (#1725200, Bio-Rad) on a CFX96 Real-Time PCR Detection System (Bio-Rad) with the cycling parameters 95.0˚C for 3 min, 39 cycles of 95.0˚C for 10 s, 63.5˚C for 10 s, 72.0˚C for 20 s. Melting curve analysis was performed at 95.0˚C for 10 s, and 65.0˚C to 95.0˚C at an increment of 5˚C for 5 s.

### Restriction site analysis and agarose gel electrophoresis

5 μL of the RT-qPCR reaction was digested with 1 μL FastDigest *PstI* restriction enzyme (#FD0614, ThermoFisher Scientific) for 1 h at 37˚C. Full-length amplicons and digestion products were separated on 1.5% agarose gel with ethidium bromide.

### Statistics

Statistical analysis and graphical illustrations were performed using GraphPad PRISM 8.3.1 (GraphPad Software). Statistical tests were applied using two-way ANOVA and corrected for multiple comparisons using Tukey´s test. Data are presented as mean ± standard deviation (SD). For all statistical analyses: * $p<0.05$, ns = Not significant.

## Results

### The evolution of a MUC3 cluster in Cercopithecoids and Hominoids

The human chromosome locus 7q22 contains three *MUC* genes *MUC3*, *MUC12*, and *MUC17*, arranged in a MUC3 cluster flanked by *ACHE* at its 5' end, and *TRIM56* and *SERPINE1* at its 3' end (Fig 1A). Using *ACHE*, *TRIM56*, and *SERPINE1* as genomic markers, we identified the MUC3 cluster in species belonging to the Catarrhini parvorder, namely Cercopithecoid (Old World monkeys) and Hominoid superfamilies, the latter including the genera *Pongo (*orangutang*)*, *Gorilla*, *Pan* (chimpanzee and bonobo) and *Homo* (Fig 1B). In Cercopithecoids we identified a MUC3 cluster with a length of 153 kilobase pairs (kbp) in *Macaca mulatta* (rhesus), while the corresponding gene cluster in the *Papio Anubis* (baboon) consisted of two mapped sequences with a total length of 138 kbp (Fig 1B). In Hominoids, MUC3 cluster length ranged from 106 kbp in the *Nomascus leucogenys* (gibbon) to 203 kbp in *Pongo abelii* (orangutang). In the Homininae subfamily, we observed striking differences between MUC3 cluster length in *Pan troglodytes* (chimpanzee) and its two closest relatives; the gene cluster in *H. sapiens* GRCh38.p13 assembly was 73 kbp shorter and in *G. gorilla* (gorilla) 66 kbp shorter than in the

chimpanzee (Fig 1B). Thus, we hypothesized that the MUC3 cluster within the human GRCh38.p13 assembly contains significant sequence gaps that may obscure unknown *MUC* genes.

To test our hypothesis, we used a set of defined criteria when exploring available primate genome assemblies for unidentified MUC3 cluster genes. We scanned the clusters for 1) start codons, 2) long mucin-type PTS-encoding exons, 3) SEA domains conserved in membrane mucins and, 4) unique intronic and exonic sequences that separate individual *MUC* genes. Our analysis revealed that all Cercopithecoids carried a MUC3 cluster consisting of *MUC3*, *MUC12*, and *MUC17* genes (Fig 1B). Strikingly, the primate *MUC3* gene existed as two distinct *MUC3A* and *MUC3B* genes, although only partial sequences of the *MUC3B* gene were identified in *P. anubis*. The Hominoid superfamily, except for *H. sapiens* and *G. gorilla*, carried a *MUC3A* gene and full or partial sequences of *MUC3B*. Thus, *we* identified a *MUC3B* gene exclusively in species belonging to the Catarrhini parvorder, which diverged from Platyrrhini (New World monkeys) around 43 million years ago (Mya) (Fig 1C). However, because of inadequate sequence coverage of the MUC3 cluster in Platyrrhini, and Scandentia (treeshrew) and Dermoptera (colugos) orders that constitute the closest relatives of primates, we were not able to determine when *MUC3B* first emerged during vertebrate mammalian evolution.

### The human 7q22 locus contains a *MUC3B* gene

Since humans and chimpanzees share 98.8% of their genomic DNA and the chimpanzee genome carries a *MUC3B* gene in the MUC3 cluster, we hypothesized that the absence of a *MUC3B* gene in humans is a result of sequence gaps in the GRCh38.p13 assembly. In the quest for a human *MUC3B* gene, we explored PacBio Single Molecule Real-Time (SMRT) reads from a human HX1 [29] and identified 3 individual reads that covered the 3' end region of *MUC3A (*encoding the C-terminal region of MUC3A protein, designated *MUC3A* C-term*)*, an intergenic region, and the 5' end region of a putative *MUC3B* gene (designated *MUC3B* N-term) (Fig 2A). Strikingly, the length of the intergenic region was on average 10,939 bp, which corresponded to the length of the *MUC3A-MUC3B* intergenic region in Catarrhines (average of 11,810 bp). Moreover, we identified 5 SMRT reads covering *MUC3B* C-term, an intergenic region, and *MUC12* N-term (Fig 2A). The average length of the *MUC3B-MUC12* intergenic region was 2469 bp and conserved in Catarrhines (average of 2491 bp). This initial exploration provided evidence for the existence of a distinct human *MUC3B* gene. However, because *MUC3A* and *MUC3B* share high sequence identity (87% and 94% for N-term and C-term across catarrhines) and the error rate of the SMRT reads was 70–85%, the HX1 assembly could not with high confidence distinguish between the two *MUC3* genes. Moreover, the reads failed to capture the length and sequence of a predicted single PTS-encoding exon in *MUC3B*.

The current GRCh38.p13 draft covers lightly packed euchromatic regions corresponding to 92% of the human genome, while more complex regions including long tandem repeats in *MUC* genes are underrepresented. A recently published CHM13 T2T v1.1 assembly, based on long-read genome sequencing of homozygous complete hydatidiform mole (CHM) cells followed by gapless telomere-to-telomere assembly, adds approximately 200 Mbp to the GRCh38.p13 assembly [12]. Importantly, the T2T-CHM13 assembly filled a 60 kbp gap between *MUC3A* and *MUC12* at locus 7q22 (Fig 2B). Within this gap, we identified a 39,267 bp long PTS-encoding exon flanked upstream by a 2,187 bp long sequence with 87% identity to *MUC3A* N-term. Downstream of the PTS-encoding exon, we identified a 6,303 bp long sequence that was 92% identical to *MUC3A* C-term and contained a SEA domain, a transmembrane domain, and a cytoplasmic tail with a conserved PDZ motif [30] (Fig 2C). Thus, our findings suggest that the T2T-CHM13 assembly contains a putative *MUC3B* gene at locus

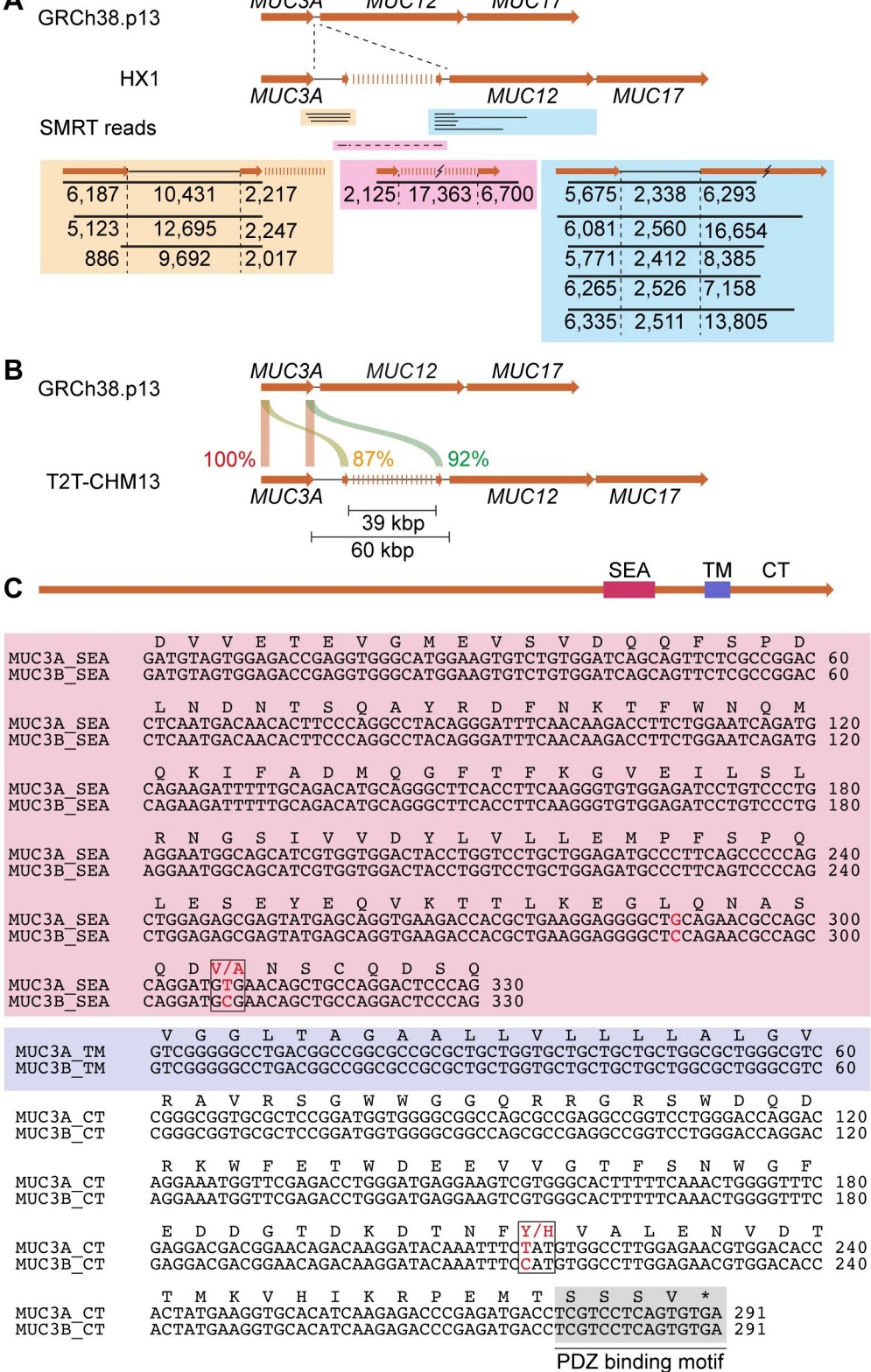

**Fig 2. Evidence of a putative *MUC3B* gene in recent human genome assemblies.** (A) Exploration of PacBio sequencing of HX1 genome identified SMRT reads covering the intergenic region between *MUC3A* and putative *MUC3B*, an incomplete PTS sequence, and intergenic sequences between putative *MUC3B* and *MUC12*. (B) The

T2T-CHM13 assembly contains a 60 kb gap between *MUC3A* and *MUC12*. (C) Sequence alignments of SEA, transmembrane (TM), and cytoplasmic tails (CT) of *MUC3A* and putative *MUC3B* show high sequence identity, nucleotide mismatches, and a conserved PDZ binding motif.

7q22 with a high sequence identity with *MUC3A*. Although previous studies have proposed the existence of a human *MUC3B* gene [31, 32], most recently in the African pan-genome [33] that contains a 22,827 bp contig aligning with *MUC3B* exon 2 (S1A Fig in S1 File), the complete length, sequence and exon-intron architecture of *MUC3B*, including its repetitive PTS-coding exon 2, remain unresolved.

## Distinct human *MUC3A* and *MUC3B* genes share high sequence homology

To better characterize the putative human *MUC3B* gene, we compared the exon-intron architecture of *MUC3B* to *MUC3A*. *MUC3A* has been reported to contain 11 exons, including a PTS-encoding exon with a length of at least 6 kbp [34]. Our analysis showed that *MUC3A* and *MUC3B* both have 12 exons, revealing a previously overlooked exon 4 (S1B, S1C Fig in S1 File). Exons of the two *MUC3* genes have nearly identical nucleotide lengths, except the single PTS-encoding exon 2 which measures 15,873 bp (5,291 amino acids) in *MUC3A* and 39,267 bp (13,089 amino acids) in *MUC3B* (Fig 3A). Nucleotide sequence identity between *MUC3A* and *MUC3B* was on average 93% for exons, and 92% for introns. The superfamilies of Hominoids (apes and humans) and Cercopithecoids (Old World monkeys) diverged around 29 Mya [6]. Sequence alignments between N- and C-termini of *MUC* genes in the MUC3 cluster showed a high degree of conservation between *H. sapiens* and members of the Cercopithecoid and Hominoid branches. Human *MUC3A* N-term was 99% identical to chimpanzee *MUC3A* N-term and 90–91% identical to *MUC3A* N-term in Cercopithecoid members rhesus and baboon. *MUC3A* C-term showed a slightly higher degree of divergence compared to *MUC3A* N-term (Fig 3B and S2 Table). The same trend was observed for *MUC3B*, in which the *MUC3B* C-term was less conserved than *MUC3B* N-term. Tandem repeat regions are prone to duplications and deletions caused by recombination [35]. Accordingly, we observed higher evolutionary sequence divergence in the PTS-encoding exon 2 of *MUC* genes in the MUC3 cluster (Fig 3B and S2 Table). Moreover, pairwise alignment of Catarrhini MUC3 cluster genes revealed a general trend toward the expansion of tandem repeats during primate evolution (S1D, S1E Fig in S1 File). Specifically, within exon 2 of human *MUC3A* and *MUC3B*, we identified imperfect repeats with 87% amino acid sequence identity between *MUC3A* and *MUC3B*. In addition, *MUC3B* contained an additional 1368 amino acids of imperfect repeats (Fig 3C). *MUC3A* and *MUC3B* also harbored 166 and 549 perfect tandem repeats, respectively, consisting of a 17 amino acids long consensus sequence (ITTTETTSHSTPSFTSS) (Fig 3D). We conclude that the genetic structure of *MUC3B* is highly similar to *MUC3A* and that the two *MUC3* genes are likely paralogous genes characterized by variable number of tandem repeats.

## *MUC3A* and *MUC3B* are regulated by conserved regulatory elements

Sequences upstream of transcription start sites (TSS) contain regulatory elements that dictate gene expression. Promotor activity within positions -1 –-242 upstream of *MUC3A* TSS has been reported previously [32], yet evolutionary conservation of the promotor regions and transcription factors that potentially regulate *MUC3* genes are largely unknown. Sequence analysis of presumed regulatory sequences -1 kbp upstream of human *MUC3A* TSS identified a candidate cis-Regulatory Element (cCRE) at position -1 –-403 bp (Fig 4A), which shared 83% identity with the corresponding region in *MUC3B*. Published DNase I hypersensitive site sequencing (DNase-seq) data sets from the human small intestine and colon revealed high

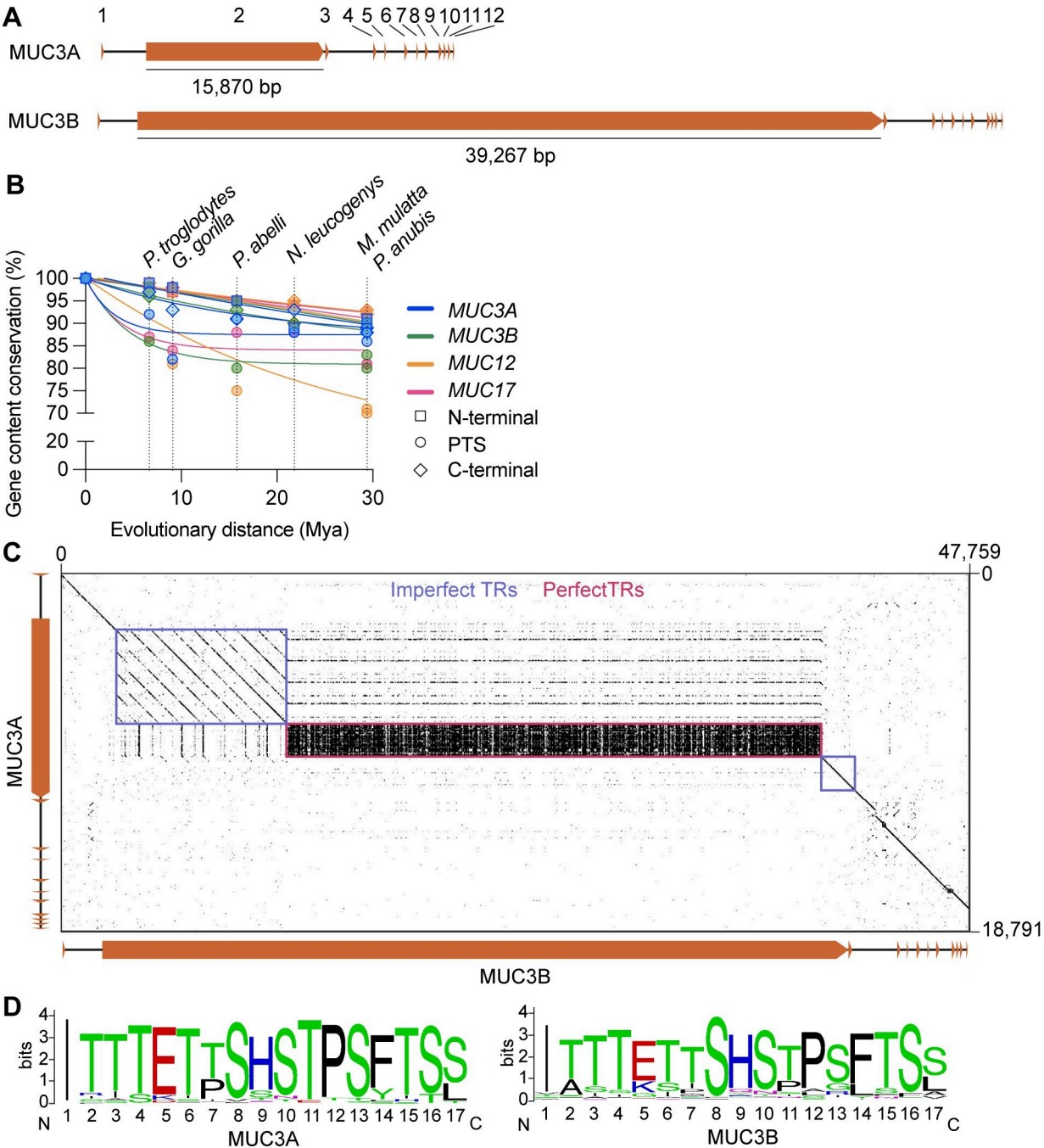

**Fig 3. Comparison of genetic and structural features of *MUC3A* and *MUC3B* genes.** (A) Exon structure and length of exon 2 of *MUC3A* and *MUC3B*. (B) The evolutionary rate of N-terminal-, PTS- and C-terminal-encoding exons in *MUC3A*, *MUC3B*, *MUC12*, and *MUC17* measured as gene content conservation (%) versus evolutionary distance (Mya). (C) Dot plot of pairwise sequence alignment of *MUC3A* and *MUC3B* identified imperfect (blue) and perfect (red) tandem repeat sequences in exon 2. (D) Sequence logo representation of 17 amino acids long consensus sequence in 166 and 549 perfect tandem repeats (TRs) in exon 2 of *MUC3A* and *MUC3B*, respectively.

signals within the cCRE (Fig 4A). Moreover, we identified high signals for active chromatin markers H3K9ac and H3K4me3 within the *MUC3A* cCRE in the human small intestine and colon, while active chromatin signals in the stomach were either low or not detected. By

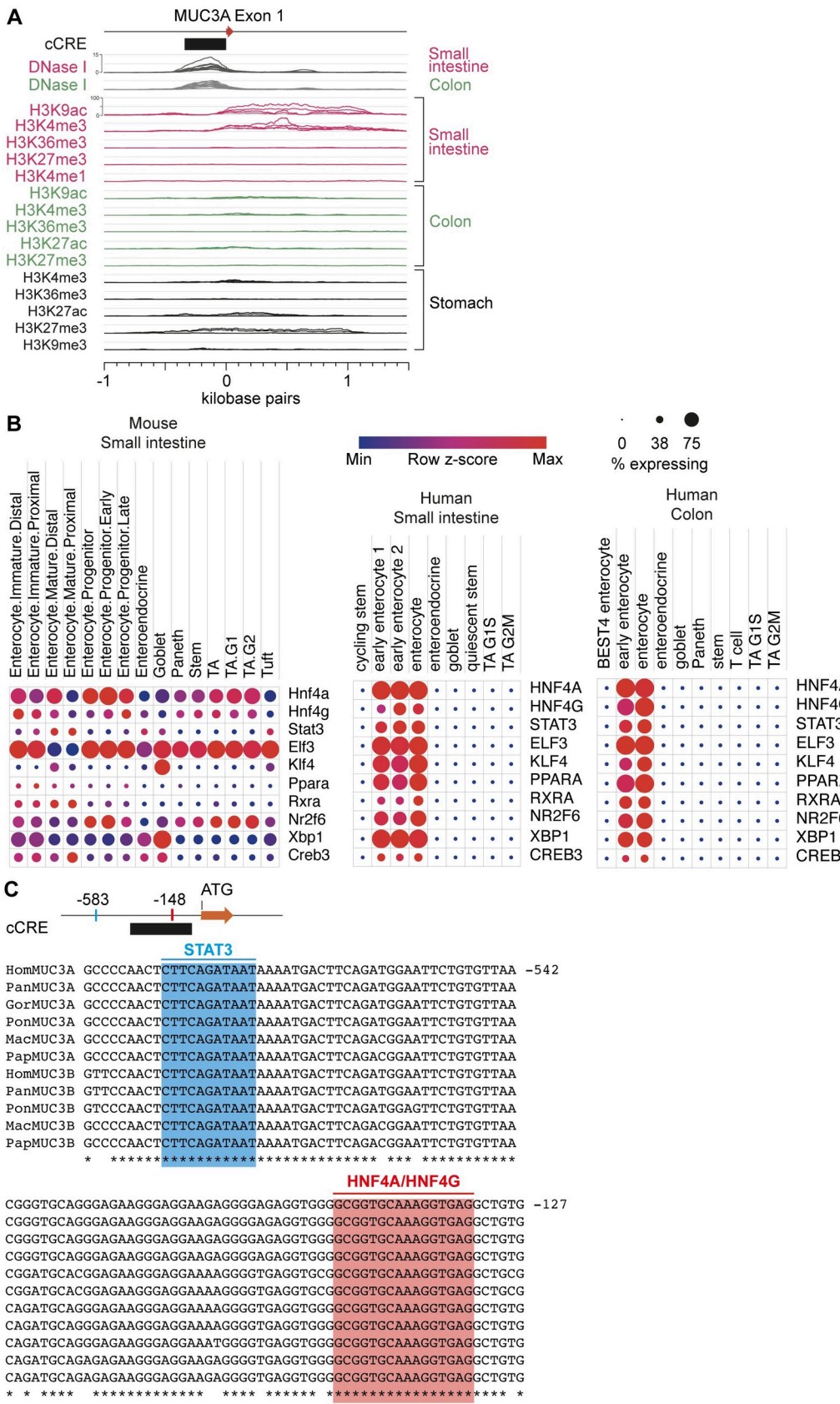

**Fig 4. Conserved regulatory elements upstream of *MUC3A* and *MUC3B* genes.** (A) Epigenetic analysis of the human small intestine and colon reveals a DNase I-sensitive cCRE and specific histone modifications surrounding the *MUC3A* transcription start site. (B) Single-cell analysis of human and mouse intestines shows gene expression of transcription factors in transporting IECs, with conserved binding sites upstream of *MUC3A* and *MUC3B*. (C) Binding sites for transcription factors STAT3 and HNF4A/G are completely conserved upstream of *MUC3A* and *MUC3B* in Cercopithecoid and Hominoid superfamilies.

predicting transcription factor binding sites (TFBSs) using the JASPAR CORE vertebrate collection [36, 37], we identified putative TFBSs in cCRE of *MUC3A* (S2 Fig in S1 File). Seven of these transcription factors (ELF3, HNF4A, HNF4G, KLF4, PPARA, STAT3, and XBP1) were enriched in transporting IECs in human and mouse intestines (Fig 4B and S3 Fig in S1 File). Alignment of putative promoter regions upstream of *MUC3A* and *MUC3B* genes in Hominoid and Cercopithecoid superfamilies identified conserved TFBSs for HNF4A, HNF4G, and STAT3, strongly suggesting that the two *MUC3* genes share an evolutionarily conserved regulatory expression program in the small intestine and colon (Fig 4C, S4 Fig in S1 File and S3 Table).

## Expression of human *MUC3A* and *MUC3B* genes in the human intestine

To determine whether *MUC3B* is transcribed into messenger RNA, we mapped published RNA-sequencing data sets from the human intestine [26], liver [27], and kidneys [38] to the T2T-CHM13 assembly. A considerable number of sequenced reads from *MUC3A* and *MUC3B* transcripts were detected in the human ileum, colon, and rectum (Fig 5A), while the liver and kidneys were devoid of transcripts from the MUC3 cluster genes (S4 Table). Our findings are supported by the human cell atlas [39], which shows that *MUC3A* is mainly expressed in epithelial cells of the small intestine and colon (1446 of 2316 $MUC3A^+$ cells) (S5 Fig in S1 File). Our data contradict a previous observation of *MUC3A* transcripts in the heart, liver, prostate, and thymus, and *MUC3B* transcripts in the small intestine and colon [40]. Notably, primer BLAST analysis showed that the *MUC3B* probe used in the study was specific for exon 3 of *MUC17*, whereas the *MUC3A* probe was specific for *MUC3A* exon 2 (S4 Table). Since these exons encode the repetitive mucin-type PTS-domain, we cannot exclude unspecific detection of mucin transcripts in other tissues.

Because exons encoding the N-terminal, PTS, and C-terminal regions of *MUC3A* and *MUC3B* share 87%, 83%, and 92% identity and PTS-encoding exons are highly repetitive, it is challenging to detect unique reads that accurately distinguish between the two *MUC3* genes. Therefore, we turned our attention to reads that map to exons 3–12, where we identified an average of 3.5±1.6 unique reads per kilobase transcript (RPK) of *MUC3A* and 13.0±4.4 unique RPK of *MUC3B* (Fig 5A and S4 Table). We next used unique and shared reads in the C-terminal region to calculate normalized gene expression of *MUC3A*, *MUC3B*, *MUC12*, and *MUC17* in the human intestine. In the ileum, *MUC17* showed significantly higher expression than *MUC3A*, *MUC3B*, and *MUC12*, while *MUC12* showed a trend towards higher expression in the rectum compared to the ileum. We detected comparable numbers of *MUC3A* and *MUC3B* transcripts in all three intestinal segments (Fig 5B).

Next, we applied targeted reverse transcriptase quantitative polymerase chain reaction (RT-qPCR) to validate the presence of unique *MUC3A* and *MUC3B* transcripts in ileum collected from five human patients (S5 Table). For this purpose, we designed gene-specific primer pairs that target exons 3 and 8 with a 9.5–12.0% mismatch between *MUC3A* and *MUC3B*. The resulting 646 bp cDNA amplicons from each gene transcript were further distinguishable by a unique *PstI* restriction site in the *MUC3A* cDNA amplicon (Fig 5C). RT-qPCR from all five patients resulted in the expected 646 bp cDNA amplicon and subsequent *PstI*-digestion

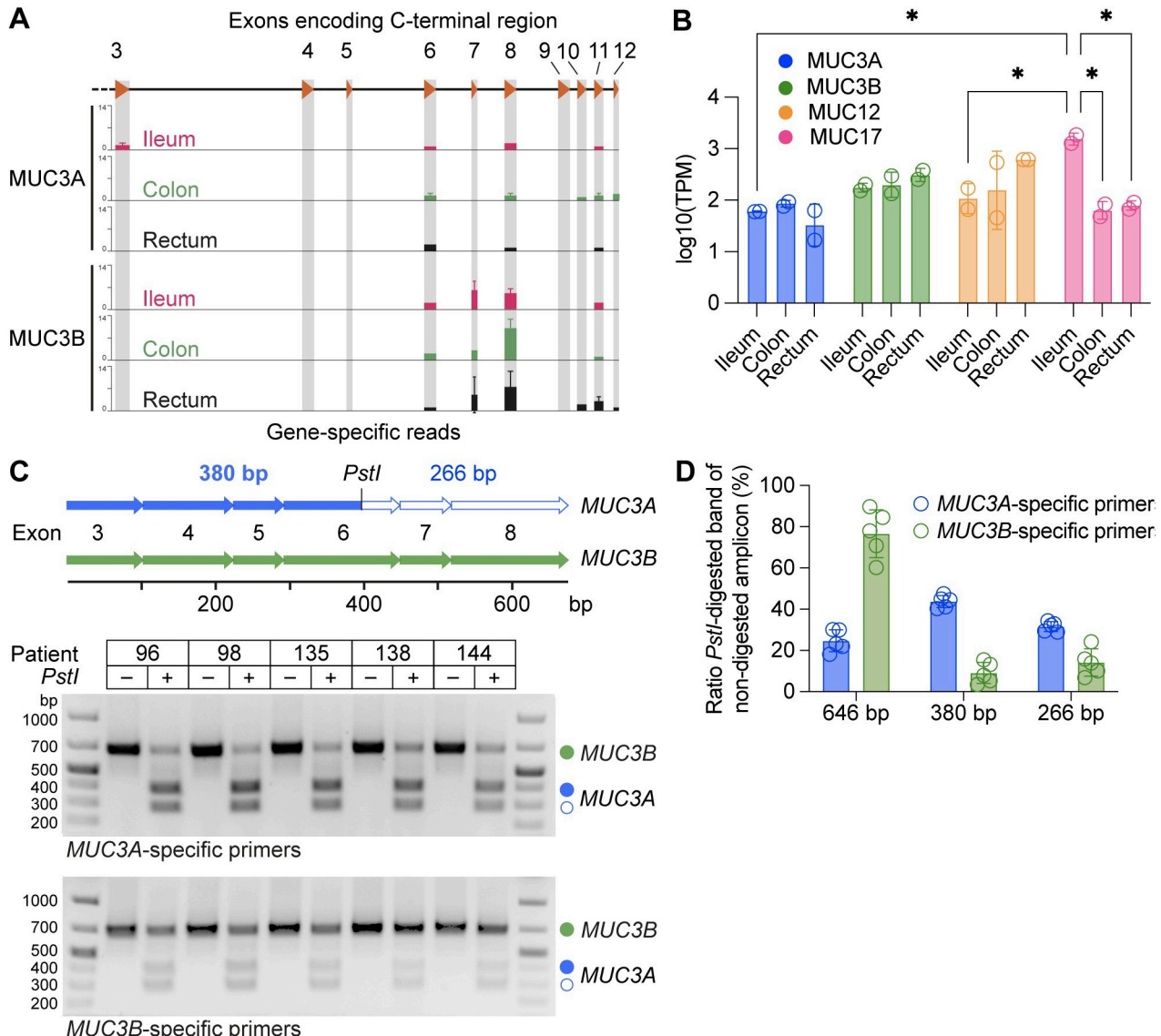

**Fig 5. Expression of *MUC3B* gene in the human intestine.** (A) Unique reads for *MUC3A* and *MUC3B* in RNA-sequencing data from human ileum, colon, and rectum mapped to T2T-CHM13 human genome assembly. (B) Gene expression of MUC3 cluster genes in human ileum, colon, and rectum. 2 samples per tissue segment. * $p < 0.05$ as determined by two-way ANOVA, corrected for multiple comparisons using Tukey´s test. Data are presented as mean ± standard deviation (SD). (C) Specific primers amplify a 646 bp cDNA spanning exons 3–8 in *MUC3A* and *MUC3B* transcripts from the ileum of five individuals. *MUC3A* cDNA carries a *PstI* restriction site in exon 6 that distinguishes *MUC3A* from *MUC3B* transcripts. Agarose gel electrophoresis of *PstI* restriction digests of amplified cDNA from *MUC3A* and *MUC3B* transcripts results in 380 bp and 266 bp fragments from *MUC3A* cDNA. (D) Quantification of bands from agarose gel in C. n = 5 individuals. Data are presented as mean ± standard deviation (SD).

produced 380 bp and 266 bp restriction fragments (Fig 5C). Notably, we observed significant differences in intensities of *PstI*-sensitive and *PstI*-resistant fragments produced by the two gene-specific primer pairs. 75% of amplicons generated by *MUC3A*-specific primers were *PstI*-sensitive and therefore originated from *MUC3A* transcripts (Fig 5D). Similarly, 76% of amplicons generated by *MUC3B*-specific primers were *PstI*-resistant *MUC3B* transcripts. Thus, despite significant sequence similarity between the two *MUC3* genes, we successfully identified and distinguished between *MUC3A* and *MUC3B* transcripts in the human intestine.

## Completion of a gapless human MUC3 cluster at locus 7q22

Finally, based on the T2T-CHM13 assembly, we revised the gapless length and sequence of all four membrane mucins genes in the MUC3 cluster at locus 7q22. In this new assembly, we identified a longer PTS-encoding exon 2 of 15,870 bp in *MUC3A* compared to 8805 bp in GRCh38.p13. The PTS-encoding exon of human *MUC12* was 32,428 bp long compared to 14,935 bp in GRCh38.p13 (S6 Table). The complete gapless sequences of *MUC3A*, *MUC3B*, *MUC12*, and *MUC17* genes at the 7q22 locus are publicly available via Mucin Biology Groups' Mucin database (http://www.medkem.gu.se/mucinbiology/databases/index.html).

## Discussion

Mucin genes contain long protein-coding sequences consisting of tandem repeats that are difficult to read and measure. As a result, many human mucin gene sequences remain incomplete. Sequence gaps also appear in the genome of *Mus musculus*, an important model organism for understanding human gene function. In an attempt to fill critical knowledge gaps in mucin genetics, we focused on a cluster of membrane mucins genes, the MUC3 cluster, at locus q22 on human chromosome 7. The MUC3 cluster is conserved in the Cercopithecoid and Hominoid superfamilies, where two distinct *MUC3A* and *MUC3B* genes are annotated in all species except in *H. sapiens* and *G. gorilla*. In this study, we leveraged the recent T2T-CHM13 assembly of the human genome to fill a 60 kb sequence gap sandwiched between *MUC3A* and *MUC12* genes. Sequence alignment revealed a membrane mucin gene that shares high structural and sequence similarity with *MUC3A*; it consists of 12 exons and carries a PTS-encoding exon 2 encompassing imperfect and perfect tandem repeats that are conserved in *MUC3A*. Moreover, the *MUC3B* gene encodes a SEA domain, a transmembrane domain, and a cytoplasmic tail with a Class I PDZ motif that is conserved in the annotated membrane mucins of the MUC3 cluster. Importantly, nucleotide mismatches in introns and exons clearly distinguished *MUC3B* from *MUC3A*. Also, the lengths of intergenic regions spanning *MUC3A*, the putative *MUC3B*, and *MUC12* corresponded to the intergenic lengths observed within the MUC3 cluster of Cercopithecoids and Hominoids. Our study presents the complete, gap-less sequence of *MUC3B* and the entire human MUC3 cluster including all introns and exons.

The evolutionary conservation of *MUC3A* and *MUC3B* genes suggests that their regulation is conserved in higher mammals. Comparative sequence alignments and available DNase I- and ChIP-seq data sets uncovered a conserved cis-regulatory element upstream of *MUC3B* that contains binding sites for transcription factors HNF4A, HNF4G, and STAT3. Notably, HNF4A and HNF4G regulate the expression of genes encoding proteins that regulate the assembly and maintenance of the microvillus-studded apical brush border in transporting IECs [41]. STAT3 acts downstream of the heteromeric epithelial cell receptor for cytokine IL-22 that regulates the expression of MUC17, which builds a protective glycocalyx barrier atop the brush border of transporting IECs [42]. Finally, mapping of published RNA-seq data sets to the T2T-CHM13 assembly identified unique sequencing reads for *MUC3A* and *MUC3B* genes in the human intestine, while gene expression was absent in the liver and kidneys. Finally, we validated high-throughput expression data by a targeted quantitative detection of distinct *MUC3A* and *MUC3B* transcripts in the human ileum. Collectively, we identified a previously unannotated *MUC3B* gene at locus 7q22 and provide evidence for its expression in human IECs.

All examined species of the Catarrhini parvorder carry *MUC3A* and *MUC3B*, while Platyrrhini and the closest evolutionary relatives of primates only carry *MUC3A*. Albeit tempting to suggest a *MUC3* gene duplication event in the Simian infraorder, the lack of long sequencing

reads (30–40 kbp) covering the MUC3 cluster in genomes outside the Catarrhini limits our understanding of when *MUC3B* emerged during evolution. Interestingly, the N- and C-terminal regions of *MUC3A* and *MUC3B* are highly conserved within Catarrhini, whereas the PTS-encoding exons exhibit higher evolutionary divergence. The PTS domains of membrane mucins genes are encoded by short nucleotide sequences organized in tandem repeats. PTS domains are generally poorly conserved and polymorphic [3] since individual repeats are added or removed through recombination to generate VNTRs. In analogy with other genes carrying VNTRs [43], our study shows that the tandem repeat regions of MUC3 cluster genes have undergone expansion during primate evolution. Low conservation and considerable polymorphism between and within species suggest that O-glycosylation of mucin VNTRs is a non-template-driven process under evolutionary and environmental pressure. For example, glycosylation of mucin VNTRs in microbe-rich environments such as the oral cavity and gastrointestinal tract are likely under selective pressure to maintain appropriate interactions with microorganisms that have coevolved with the host through various periods of geographical, dietary, and lifestyle adaptations. This co-speciation is evident in the gastrointestinal tract, where the microbiome of present-day humans is enriched in mucin-degrading genes compared to a higher abundance of starch- and chitin-degrading genes in our ancestral microbiome [44]. Another example is found in the epithelial cell surface glycocalyx, where O-glycosylation underwent major remodeling >2 Mya when a human ancestor acquired an inactivating mutation in *CMAH*, a gene responsible for converting N-acetylneuraminic acid (Neu5Ac) to N-glycolylneuraminic acid (Neu5Gc) [45]. The resulting accumulation of terminal Neu5Ac in the glycocalyx of human cells has since been exploited by numerous pathogens such as *Vibrio cholera* [46] and SARS-CoV-2 [47]. The emergence of mucin genes as a result of environmental adaptation has been attributed to the process of convergent evolution, where genes encoding proline-rich proteins independently gain serine and threonine residues that assemble into tandemly repeated O-glycosylated mucin domains [48].

Due to existing challenges in sequencing very long repetitive regions, the nature of mucin polymorphism and its contribution to human disease phenotypes remains elusive. A recent study showed that the length of VNTRs in membrane mucin MUC1 is associated with several disease phenotypes related to kidney function [49], supporting the notion that glycosylated PTS domains of membrane mucins play critical roles in organ function and homeostasis. Intestinal membrane mucin MUC17 is genetically and structurally related to MUC3A and MUC3B and functions as a major building block of the dense glycocalyx covering transporting IECs. In mouse small intestine, Muc17 expression is induced during the suckling-weaning transition when the quantity and complexity of the gut microbiota increases and creates a demand for IECs to establish a cell-attached glycocalyx that prevents adhesion of luminal bacteria to the epithelium [42]. While the function of the *MUC3A*, *MUC3B*, and *MUC12* remains elusive, their expression varies along different segments of the human intestine, suggesting that MUC3 cluster genes perform segment- and cell-specific functions in humans and other mammalian vertebrates. Our comprehensive map of the MUC3 cluster in the human genome provides opportunities to identify new VNTR polymorphisms associated with disease phenotypes and allows for future exploration of gene orthologs of the MUC3 cluster in experimental mammalian models such as the mouse.

## Supporting information

**S1 Table. DNase-sequencing and Chromatin immunoprecipitation sequencing data sets used in this study.**
(XLSX)

**S2 Table. Conservation of N-terminal, PTS, and C-terminal regions of MUC3 cluster genes in primates.**
(XLSX)

**S3 Table. Binding sites upstream of the *MUC3A* gene transcription start site for transcription factors enriched in intestinal epithelial cells.**
(XLSX)

**S4 Table. Identification of MUC3 cluster genes in human tissues.**
(XLSX)

**S5 Table. Patient demographics.**
(XLSX)

**S6 Table. Statistical summary of MUC3 cluster genes belonging to members of Cercopithecoids and Hominoids superfamilies.**
(XLSX)

**S1 File. Supporting information containing S1-S5 Figs and supplementary methods.**
(DOCX)

**S2 File. Raw image of agarose gel shown in Fig 5C.**
(PDF)

## Acknowledgments

We thank Professor Gunnar C. Hansson for the valuable discussions.

## Author Contributions

**Conceptualization:** Thaher Pelaseyed.

**Data curation:** Tiange Lang, Thaher Pelaseyed.

**Formal analysis:** Tiange Lang, Thaher Pelaseyed.

**Funding acquisition:** Thaher Pelaseyed.

**Investigation:** Tiange Lang, Thaher Pelaseyed.

**Methodology:** Tiange Lang, Thaher Pelaseyed.

**Project administration:** Thaher Pelaseyed.

**Resources:** Thaher Pelaseyed.

**Software:** Tiange Lang, Thaher Pelaseyed.

**Supervision:** Thaher Pelaseyed.

**Validation:** Thaher Pelaseyed.

**Visualization:** Thaher Pelaseyed.

**Writing – original draft:** Thaher Pelaseyed.

**Writing – review & editing:** Tiange Lang, Thaher Pelaseyed.

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
