## [Decision Letter · Decision Letter 0]

29 Jun 2022

PONE-D-22-00623Discovery of a *MUC3B* gene reconstructs the membrane mucin gene cluster on human chromosome 7PLOS ONE

Dear Dr. Pelaseyed,

Thank you for submitting your manuscript to PLOS ONE. After careful consideration, we feel that it has merit but does not fully meet PLOS ONE’s publication criteria as it currently stands. Therefore, we invite you to submit a revised version of the manuscript that addresses the points raised during the review process. Please pay particular attention to the comments from the reviewers concerning additional discussion points. especially those associated with currently non-discussed studies.

We look forward to receiving your revised manuscript.

Kind regards,

Michael Scott Brewer, Ph.D.

Academic Editor

PLOS ONE

Journal Requirements:

In your cover letter, please note whether your blot/gel image data are in Supporting Information or posted at a public data repository, provide the repository URL if relevant, and provide specific details as to which raw blot/gel images, if any, are not available. Email us at plosone@plos.org if you have any questions

"We thank Professor Gunnar C. Hansson for valuable discussions. This work was supported by the Swedish Society for Medical Research (Svenska Sällskapet för Medicinsk Forskning, grant S17-0005), National Institutes of Health (grants 5U01AI095542-08-WU-19-95 and 5U01AI095542-09-WU-20-77), Wenner-Gren Foundations (grants FT2017-0002, UPD2018-0065, and WUP2017-0005), Jeansson Foundations (grant JS2017-0003), and the Åke Wiberg Foundation (grant M17-0062)."

"TP was supported by

- Grant S17-0005, Swedish Society for Medical Research, https://www.ssmf.se

- Grants 5U01AI095542-08-WU-19-95 and 5U01AI095542-09-WU-20-77, National Institutes of Health, https://www.niaid.nih.gov

- Grants FT2017-0002, UPD2018-0065, and WUP2017-0005, Wenner-Gren Foundations, https://www.swgc.org/

- Grant JS2017-0003, Jeansson Foundations, http://jeanssonsstiftelser.se/en/

- Grant M17-0062, Åke Wiberg Foundation, https://ake-wiberg.se/

Reviewers' comments:

Reviewer's Responses to Questions

**Comments to the Author**

1. Is the manuscript technically sound, and do the data support the conclusions?

Reviewer #1: Yes

Reviewer #2: Yes

2. Has the statistical analysis been performed appropriately and rigorously? 

Reviewer #1: Yes

Reviewer #2: Yes

3. Have the authors made all data underlying the findings in their manuscript fully available?

Reviewer #1: Yes

Reviewer #2: Yes

4. Is the manuscript presented in an intelligible fashion and written in standard English?

Reviewer #1: Yes

Reviewer #2: Yes

5. Review Comments to the Author

Reviewer #1: This study looks to resolve a complex locus of the human genome that contains a cluster of transmembrane mucin genes. Since mucin genes are rich in repeats, short read sequencing technologies have trouble genotyping mucins, and can even lead to misassemblies or gaps in the genome. The authors focus on resolving this specific gene locus by analyzing recently available human genome assembly, CH1M3. By integrating cross-primate syntenic analysis, available tissue expression data, and validation approaches, the authors confirm the presence of MUC3B in the transmembrane mucin cluster. Overall the authors findings help resolve a complicated locus in the human genome.

I think the paper adds valuable data to the mucin genetics community. I have some comments that should be addressed.

Major Comments:

Although the authors are first to better resolve this gene locus using the new T2T human reference genome in an independent study, the authors only briefly stated that MUC3B has been previously found and distinguished from MUC3A : https://www.sciencedirect.com/science/article/abs/pii/S0006291X00934065?via%3Dihub, https://www.jbc.org/article/S0021-9258(20)75740-6/fulltext. Recent papers have highlighted this locus as having MUC3-like absence in the hg38 reference genome: https://www.nature.com/articles/s41588-018-0273-y.pdf. These papers should be discussed further as they described MUC3B at the sequence and transcriptomic levels previously. Overall more emphasis should be placed on prior discovery and distinguish the findings of this particular study from previous work

In Line 188 the authors state that the MUC3B “exclusively” evolved in Catarrhini? Was there any sort of BLAST analysis, or syntenic analysis in other species where MUC3B may be ancestral or have evolved recurrently in other species. If not, the language should be toned-down and stated that other species outside of primates were not examined.

Minor Comments:

The author should assess the language more carefully in the manuscript. There are several typos and grammatical issues, particularly in the Discussion section. The examples include but are not limited to the following:

Line 38 : “characterized” is used twice

Line 49: “investigation of” should read “to investigate”

Line 63: “reach” should read “reaches”

Line 229: “in” should be “on”

Line 271: should the second “MUC3A” in the sentence read “MUC3B”?

Line 308: The opening sentence of the Discussion is not a complete sentence.

Line 309: “As results” should read “as a result”

Line 310: “uncomplete” should read “incomplete”

In the introduction lines 69-81 the authors describe the MUC3 sequence of the mouse as having homology to MUC17. There is no paper cited for this claim.

Reviewer #2: The manuscript entitled “Discovery of a MUC3B gene reconstructs the membrane mucin gene cluster on human chromosome 7” used patients’ tissue from colonoscopy to construct sequence alignment and sequence based on VNTRs in human genome assemblies. The study design was well with valid verification. However, is the novel gene reconstructs specific for intestinal tissues? Or general expression? Please discuss this part in the discussion part.

6. PLOS authors have the option to publish the peer review history of their article (what does this mean?). If published, this will include your full peer review and any attached files.

Reviewer #1: No

Reviewer #2: No

---

## [Author Response · Author response to Decision Letter 0]

23 Aug 2022

Response Reviewers (see “Revised Manuscript with Track Changes”)

Reviewer #1

Major Comments:

1. Although the authors are first to better resolve this gene locus using the new T2T human reference genome in an independent study, the authors only briefly stated that MUC3B has been previously found and distinguished from MUC3A:

https://www.sciencedirect.com/science/arNcle/abs/pii/S0006291X00934065?via%3Dihub

https://www.jbc.org/arNcle/S0021-9258(20)75740-6/fulltext.

Recent papers have highlighted this locus as having MUC3-like absence in the hg38 reference genome: https://www.nature.com/arNcles/s41588-018-0273-y.pdf.

These papers should be discussed further as they described MUC3B at the sequence and transcriptomic levels previously. Overall more emphasis should be placed on prior discovery and distinguish the findings of this particular study from previous work.

We thank reviewer #1 for these important comments. We made substantial additions to the original manuscript to address and discuss earlier works by: 

Gum JR et al (1997) MUC3 human intestinal mucin. Analysis of gene structure, the carboxyl terminus, and a novel upstream repetitive region. J Biol Chem 272: 26678–86

Pratt WS et al (2000) Multiple Transcripts of MUC3: Evidence for Two Genes, MUC3A and MUC3B. Biochem Biophys Res Commun 275: 916–923

Gum JR et al (2003) Initiation of transcription of the MUC3A human intestinal mucin from a TATA-less promoter and comparison with the MUC3B amino terminus. J Biol Chem 278: 49600–49609

Sherman RM et al (2018) Assembly of a pan-genome from deep sequencing of 910 humans of African descent. Nat Genet 2018 511 51: 30–35

On lines 355-359, we discuss work by Sherman et al (2018). Our sequence alignment shows that contig 316 aligns with exon 2 of MUC3B in T2T-CHM13. Our alignment also reveals that contig 316 contains synonymous, missense, and nonsense SNPs as well as in-frame insertions and deletions. The alignment is presented in S1A Fig.

On lines 363-366, we discuss work by Gum et al (1997). The study identified the MUC3 gene (later named MUC3A), encoded by 11 exons. Pratt et al (2000) also identified 11 exons for MUC3A. Our study updates the exon-intron architecture of MUC3A and MUC3B, both of which consists of 12 exons. Importantly, we identified a new exon 4 that was overlooked in the earlier publications. We have added this information in S1B-C Fig.

On lines 427-429, we acknowledge work by Gum et al (2003), in which the MUC3A promotor region was mapped to positions -1 - -242 upstream of the transcription start site. Our study complements these findings by adding epigenetic and evolutionary analysis of the promotor regions of MUC3A and MUC3B in primates.

2. In Line 188 the authors state that the MUC3B “exclusively” evolved in Catarrhini? Was there any sort of BLAST analysis, or syntenic analysis in other species where MUC3B may be ancestral or have evolved recurrently in other species. If not, the language should be toned-down and stated that other species outside of primates were not examined.

We thank reviewer #1 for the valuable comment. We have used BLAST analysis to search for MUC3B in species belonging to New World monkeys (Platyrrhini), as well as Scandentia (treeshrew) and Dermoptera (colugos) orders, which share a common ancestor with primates. While we find partial or complete sequences for MUC3A, MUC12, and MUC17 in these branches of life, we have not been able to identify any unique sequences belonging to MUC3B outside the Catarrhini parvorder. The available genomic sequences for the species outside Catarrhini lack long reads that cover the MUC3 cluster, hampering our efforts to distinguish between MUC3A and MUC3B. We conclude that available genomes outsides Catarrhini suffer from gaps, as has been observed for human genome assemblies prior to T2T-CHM13. Consequently, we have not been able to determine when during vertebrate mammalian evolution a gene duplication event gave rise to two distinct MUC3 genes. 

We have toned down the language on lines 30-32. We have added the above conclusions on lines 288-294 and lines 590-594.

Minor Comments:

1. The author should assess the language more carefully in the manuscript. There are several typos and grammatical issues, particularly in the Discussion section. The examples include but are not limited to the following:

Line 38 : “characterized” is used twice

Line 49: “investigation of” should read “to investigate”

Line 63: “reach” should read “reaches”

Line 229: “in” should be “on”

Line 271: should the second “MUC3A” in the sentence read “MUC3B”?

Line 308: The opening sentence of the Discussion is not a complete sentence.

Line 309: “As results” should read “as a result”

Line 310: “uncomplete” should read “incomplete”

We have corrected all the above mistakes kindly pointed out by the reviewer. In addition, we have carefully assessed the language and corrected the grammar and typos.

2. In the introduction lines 69-81 the authors describe the MUC3 sequence of the mouse as having homology to MUC17. There is no paper cited for this claim.

We apologize for overlooking the reference for this statement. We have added the correct reference in line 100.

Reviewer #2

The manuscript entitled “Discovery of a MUC3B gene reconstructs the membrane mucin gene cluster on human chromosome 7” used patients’ tissue from colonoscopy to construct sequence alignment and sequence based on VNTRs in human genome assemblies. The study design was well with valid verification. However, is the novel gene reconstructs specific for intestinal tissues? Or general expression? Please discuss this part in the discussion part.

We thank reviewer #2 for the valuable comment. In the original version of our manuscript, we analyzed single-cell RNA-seq from the human ileum, colon, and rectum as well as kidneys and liver. We found a high number of unique reads for MUC3A, MUC3B, MUC12, and MUC17 in the ileum, colon, and rectum. None of the genes were expressed in the kidneys and liver (see S3 Table, Tab A). We further experimentally validated mucin gene expression in the human ileum. Our findings are supported by the human cell atlas Tabula sapiens. Single-cell analysis of 500,000 cells from 24 organs of 15 normal human subjects showed that >60% of MUC3A+ cells constitute various populations of intestinal epithelial cells, dominated by transporting epithelial cells (enterocytes). We have added this analysis as a new S5 Fig, and we discuss the findings on lines 478-480.

Notably, our findings contradict previous observations by Kyo K, et al ((2001) J Hum Genet 2001 461 46: 5–20), who detected MUC3A transcripts in a broad set of human tissues (heart, liver, prostate, and thymus) using Northern blot. Although we cannot fully explain this discrepancy, our primer-BLAST analysis shows that probes against MUC3B were in fact specific for the repetitive exon of MUC17. Probes against MUC3A were specific but hybridized with exon 2 of MUC3A, which is a highly repetitive region with tandem repeats occurring in all members of the mucin gene family. Thus, we can exclude unspecific hybridization to mucin tandem repeat transcripts in the heart, liver, prostate, and thymus. We have inserted our conclusions on lines 480-485 and in S3 Table (tab E).

---

## [Decision Letter · Decision Letter 1]

20 Sep 2022

PONE-D-22-00623R1Discovery of a *MUC3B* gene reconstructs the membrane mucin gene cluster on human chromosome 7PLOS ONE

Dear Dr. Pelaseyed,

Thank you for submitting your manuscript to PLOS ONE. After careful consideration, we feel that it has merit but does not fully meet PLOS ONE’s publication criteria as it currently stands. Therefore, we invite you to submit a revised version of the manuscript that addresses the points raised during the review process.

The reviewers and I are generally pleased with the revisions. Thank you for taking the time and care to address their questions. Please see the comments from the reviewer concerning the newly published material. Beyond this, I am happy to move forward. I just want to give you a chance to include the newer material.

We look forward to receiving your revised manuscript.

Kind regards,

Michael Scott Brewer, Ph.D.

Academic Editor

PLOS ONE

Journal Requirements:

Reviewers' comments:

Reviewer's Responses to Questions

**Comments to the Author**

1. If the authors have adequately addressed your comments raised in a previous round of review and you feel that this manuscript is now acceptable for publication, you may indicate that here to bypass the “Comments to the Author” section, enter your conflict of interest statement in the “Confidential to Editor” section, and submit your "Accept" recommendation.

Reviewer #1: All comments have been addressed

2. Is the manuscript technically sound, and do the data support the conclusions?

Reviewer #1: Yes

3. Has the statistical analysis been performed appropriately and rigorously? 

Reviewer #1: Yes

4. Have the authors made all data underlying the findings in their manuscript fully available?

Reviewer #1: Yes

5. Is the manuscript presented in an intelligible fashion and written in standard English?

Reviewer #1: Yes

6. Review Comments to the Author

Reviewer #1: The authors have answered my comments to my satisfaction. They have also thoroughly revised the language and grammar in the manuscript, and it reads much better. I have one additional suggestion that I recommend. During the time of this review, two highly relevant publications on mucins were published in high profile journals. It would be wonderful if the authors could cite these two papers in the discussion or introduction. The first proposes a new way mucins evolve (also shows MUC3 locus across species): https://www.science.org/doi/10.1126/sciadv.abm8757 . The other redefines the number of mucins in humans: https://www.nature.com/articles/s41467-022-31062-4.

7. PLOS authors have the option to publish the peer review history of their article (what does this mean?). If published, this will include your full peer review and any attached files.

Reviewer #1: No

---

## [Author Response · Author response to Decision Letter 1]

20 Sep 2022

Response Reviewers (see “Revised Manuscript with Track Changes”)

Reviewer #1

The authors have answered my comments to my satisfaction. They have also thoroughly revised the language and grammar in the manuscript, and it reads much better. I have one additional suggestion that I recommend. During the time of this review, two highly relevant publications on mucins were published in high profile journals. It would be wonderful if the authors could cite these two papers in the discussion or introduction. The first proposes a new way mucins evolve (also shows MUC3 locus across species): https://www.science.org/doi/10.1126/sciadv.abm8757 . The other redefines the number of mucins in humans: https://www.nature.com/articles/s41467-022-31062-4.

We thank reviewer #1 for these important comments. We have carefully read the recommended papers.

Pajic et al. (Science Advances 2022) propose convergent evolution as a mechanism by which precursor genes coding for proline-rich proteins gain serine and threonine residues that form mucin-type O-glycosylated exonic repeats. The authors suggest that the retention of novel mucins genes is beneficial for the host as mucins are mechanistic lubricants, serve as matrices enclosing immunological factors, and mediate host-microbe interactions. These arguments are also presented by us in the “Discussion” section. Thus, we have added a reference to Pajic et al. at lines 433-437.

Malaker et al. (Nature Communications 2022) used in silico and mucin-selective affinity purification strategies to identify proteins with a putative mucin domain. Expectedly, the reported “mucinome” includes known members of the canonical mucin family (MUC1, MUC4, MUC5AC, MUC5B, MUC6, MUC13, MUC16, and MUC20), whereas the study did not experimentally detect other canonical mucins (MUC2, MUC3A, MUC3B, MUC12, and MUC17) since the analyzed biological samples did not have an intestinal origin. The canonical mucin family is defined by the presence of evolutionarily conserved protein domains that together with an extended and repetitive proline-threonine-serine (PTS)-rich domain construct gel-forming and membrane mucins. The conserved domains include vWD, CysD, and CK domains in gel-forming mucins [1], and SEA or NIDO-AMOP-VWD domains in membrane mucins [2–5].

Notably, many of the detected proteins by Malaker et al. do not carry these conserved domains and exhibit lower PTS content compared to canonical mucins. See table below for the PTS content of a selection of identified proteins.

Uniprot Protein PTS (% of total residues) Canonical mucin (YES/NO)

P15941 MUC1 44.8 Yes

P98088 MUC5AC 45.6 Yes

Q9H3R2 MUC13 30.6 Yes

Q685J3 MUC17 56.5 Yes

O00468 Agrin 21.9 No

Q14118 DAG1 20.0 No

Q6WRI0 IGSF10 26.4 No

Q14114 LRP8 19.2 No

Q6ZSS7 MFSD6 21.8 No

P14543 NID1 22.3 No

Q8N131 Porimin 35.6 No

While the idea of a “mucinome” is conceptually intriguing, we argue for a more stringent definition of what constitutes a mucin [6]. Thus, we have referenced Malaker et al. since the study does not identify novel genes that fulfil the criteria for canonical mucins.

References

1. Trillo-Muyo S, Nilsson HE, Recktenwald C V., Ermund A, Ridley C, Meiss LN, et al. Granule-stored MUC5B mucins are packed by the noncovalent formation of N-terminal head-to-head tetramers. J Biol Chem. 2018;293: 5746. doi:10.1074/jbc.RA117.001014

2. Moniaux N, Nollet S, Porchet N, Degand P, Laine A, Aubert JP. Complete sequence of the human mucin MUC4: a putative cell membrane-associated mucin. Biochem J. 1999;338: 325. doi:10.1042/0264-6021:3380325

3. Moniaux N, Escande F, Porchet N, Aubert JP, Batra SK. Structural organization and classification of the human mucin genes. Front Biosci. 2001/10/02. 2001;6: D1192-206. Available: http://www.ncbi.nlm.nih.gov/pubmed/11578969

4. Ligtenberg MJL, Kruijshaar L, Buijs F, Van Meijer M, Litvinov S V., Hilkens J. Cell-associated episialin is a complex containing two proteins derived from a common precursor. J Biol Chem. 1992;267: 6171–6177. doi:10.1016/S0021-9258(18)42677-4

5. Macao B, Johansson DG, Hansson GC, Hard T. Autoproteolysis coupled to protein folding in the SEA domain of the membrane-bound MUC1 mucin. Nat Struct Mol Biol. 2006;13: 71–76. Available: http://www.ncbi.nlm.nih.gov/entrez/query.fcgi?cmd=Retrieve&db=PubMed&dopt=Citation&list_uids=16369486

6. Arike L, Hansson GC. The Densely O-glycosylated MUC2 Mucin Protects the Intestine and Provides Food for the Commensal Bacteria. J Mol Biol. 2016;428: 3221. doi:10.1016/J.JMB.2016.02.010

---

## [Editor Report · Decision Letter 2]

21 Sep 2022

Discovery of a *MUC3B* gene reconstructs the membrane mucin gene cluster on human chromosome 7

PONE-D-22-00623R2

Dear Dr. Pelaseyed,

We’re pleased to inform you that your manuscript has been judged scientifically suitable for publication and will be formally accepted for publication once it meets all outstanding technical requirements.

Kind regards,

Michael Scott Brewer, Ph.D.

Academic Editor

PLOS ONE
---

## [Editor Report · Acceptance letter]

26 Sep 2022

PONE-D-22-00623R2 

Discovery of a MUC3B gene reconstructs the membrane mucin gene cluster on human chromosome 7 

Dear Dr. Pelaseyed:

I'm pleased to inform you that your manuscript has been deemed suitable for publication in PLOS ONE. Congratulations! Your manuscript is now with our production department. 

Kind regards, 

on behalf of

Dr. Michael Scott Brewer 

Academic Editor

PLOS ONE